# Oral Health Knowledge, Attitudes and Practices of People Living with Diabetes in South Asia: A Scoping Review

**DOI:** 10.3390/ijerph192113851

**Published:** 2022-10-25

**Authors:** Prakash Poudel, Lal B. Rawal, Ariana Kong, Uday N. Yadav, Mariana S. Sousa, Biraj Karmacharya, Shaili Pradhan, Ajesh George

**Affiliations:** 1eHealth, NSW Government, St Leonards, NSW 2065, Australia; 2Australian Centre for Integration of Oral Health (ACIOH), School of Nursing & Midwifery, Western Sydney University, Liverpool, NSW 2170, Australia; 3Ingham Institute for Applied Medical Research, Liverpool, NSW 2170, Australia; 4School of Health, Medical and Applied Sciences, Central Queensland University, Sydney Campus, Sydney, NSW 2000, Australia; 5Physical Activity Research Group, Appleton Institute, Central Queensland University, Norman Gardens, QLD 4710, Australia; 6Translational Health Research Institute (THRI), Western Sydney University, Campbelltown, NSW 2560, Australia; 7National Centre for Epidemiology and Population Health, The Australian National University, Canberra, ACT 2601, Australia; 8Centre for Primary Health Care and Equity, University of New South Wales, Sydney, NSW 2052, Australia; 9Improving Palliative, Aged and Chronic Care through Clinical Research and Translation (IMPACCT), Faculty of Health, University of Technology Sydney, Broadway, NSW 2007, Australia; 10Department of Community and Public Health Dentistry, Dhulikhel Hospital, Kathmandu University School of Medical Sciences, Panauti 45209, Nepal; 11Department of Dental Surgery, National Academy of Medical Sciences, Bir Hospital, Mahabouddha, Kathmandu 44600, Nepal; 12School of Dentistry, Faculty of Medicine and Health, University of Sydney, Surry Hills, NSW 2010, Australia; 13La Trobe Rural Health School, La Trobe University, Bendigo, VIC 3552, Australia

**Keywords:** oral health, health knowledge, attitudes, practice, diabetes mellitus, health policy

## Abstract

Diabetes increases the risk of oral health complications. This review aimed to synthesise the current evidence on the oral health knowledge, attitudes and practices of people living with diabetes in South Asian countries and provide recommendations on possible strategies for addressing the gaps in oral health care in this population, including the role of non-dental professionals. Using a scoping review framework, six electronic databases (Ovid Medline, CINAHL, ProQuest Central, Scopus, Web of Science and Embase) were searched to identify the relevant literature published between January 2000 and December 2021. The data were extracted into three main categories based on the review’s aims, and further refined into sub-categories. A total of 23 studies were included. The review identified that while people with diabetes living in South Asian countries had some level of awareness about oral health and limited care practices to maintain good oral health, there were gaps in knowledge, and there were areas where their oral health practices and attitudes could be improved. The findings suggest a need for developing targeted oral health policies as well as implementing integrated oral health care interventions involving non-dental professionals to improve the oral health outcomes of people with diabetes.

## 1. Introduction

Diabetes mellitus is one of the most rapidly growing chronic conditions in the world, affecting approximately 537 million individuals worldwide [1]. Evidence shows that the burden of diabetes has been increasing substantially in low- and middle-income countries [2,3,4] and is strongly associated with being overweight, poor nutrition, sedentarism and socioeconomic inequalities [2,5,6,7]. People living in South Asian countries (including Afghanistan, Bhutan, Bangladesh, India, Pakistan, Maldives, Myanmar, Nepal and Sri Lanka) are known to have an increased predisposition for diabetes, which in recent decades has become a major public health concern, placing a substantial burden on healthcare services in this region [5,8]. The International Diabetes Federation [1] reported that the prevalence of diabetes in India is about 74.2 million whereas Pakistan has an estimated 33 million cases. These two countries in South Asia have the second and third highest prevalence of diabetes in the world, behind China [1].

Elevated blood glucose levels (hyperglycaemia) leads to several microvascular and macrovascular complications such as diabetic retinopathy; neuropathy; nephropathy, heart, peripheral arterial, and cerebrovascular disease; obesity, certain cancers; pancreatitis; erectile dysfunction; and non-alcoholic fatty liver disease [9,10]. Hyperglycaemia also leads to various oral complication and more particularly, periodontal (gum) diseases. Periodontal diseases include two stages: gingivitis and periodontitis. Gingivitis refers to the reversible inflammation of the gum (gingiva), while periodontitis is characterised by chronic inflammation where there is also irreversible destruction of the supporting structures (issue and alveolar bone) around the teeth [11,12]. There is a bidirectional relationship between hyperglycaemia and periodontitis, suggesting a common pathway of inflammation [13,14]. Hyperglycaemia can increase the risk of periodontitis whereas periodontitis can negatively affect glycaemic management and worsen diabetes-related complications [15,16]. It is estimated that individuals with diabetes have a greater risk, often three-fold, of developing periodontitis compared to non-diabetic individuals [16]. 

Oral health remains a major and poorly managed issue in South Asian countries. About 65% of rural Indian adults suffer from caries and periodontitis, which is one of the highest rates in the South Asian region [17]. Poor oral health can significantly impact on general health and quality of life. Studies in South Asia have found that poor oral health among adults increased their physical, functional limitation such as when eating, and increased psychological discomfort [18,19]. Some contributing factors to poor oral health that are common to this population are diets consisting mainly of carbohydrates with a high glycaemic index and the widespread use of tobacco [17,20]. In addition, studies have also indicated poor oral health literacy among people with diabetes [21,22] which are compounded by poor access to dental treatment, particularly among rural and underprivileged populations [23]. While these issues are complex, increasing oral health knowledge and improving attitudes among people with diabetes could be addressed through the development and implementation of oral health models of care that involve non-dental clinicians [24,25]. Various studies across the world have demonstrated that a wide range of non-dental clinicians have advocated for oral health among clients, leading to improved oral health outcomes [26,27,28]. Nevertheless, as these studies have been conducted in other countries, little is known about the receptiveness of people living in South Asian countries towards receiving oral health education from non-dental clinicians.

Despite the increased risk of oral health problems among people living with diabetes, a recent review indicated that people with diabetes have poor oral health knowledge, a lack of awareness, and compliance with certain oral health behaviours (such as brushing and flossing) and lower probability of visiting a dentist regularly [29]. Unfortunately, the current literature has had a broader focus, missing the unique barriers for people with diabetes in developing countries, including South Asia, where the absence of universal health care and insurance makes access to dental care often challenging. Studies suggest that awareness or knowledge about oral health are found to be positively associated with oral health behaviours such as increased frequency of brushing and dental visits [30] and good periodontal health status [31]. Furthermore, it is reported that oral health behaviours are influenced by the social determinants of health [32], and people who are from disadvantaged or lower socioeconomic groups often have more limited knowledge and negative attitudes to oral health and uptake of dental services, and as a result, are more likely to experience the burden of oral disease [33,34]. Since the prevalence of diabetes and dental problems are higher in South Asia, it is important to understand the distinctive gaps in knowledge, attitudes and practices of this population to help inform preventive strategies, policies and practices required to improve oral health outcomes. In this review, we aimed to synthesise current evidence on the oral health knowledge, attitudes and practices of people living with diabetes in South Asian countries and provide recommendations on possible strategies for addressing the gaps in oral health care in this population, including the role of non-dental professionals.

## 2. Materials and Methods

This scoping review used the framework outlined by Arksey and O’Malley [35] to map the existing evidence and identify gaps in the literature within this emerging field. A scoping review also provides flexibility in the quality and breadth of literature included. The scoping review process also provided an iterative, rather than linear process, which enabled the research team to go back-and-forth and redefine search terms and study aims based on initial findings [35,36]. This scoping review followed the Preferred Reporting Items for Systematic Review (PRISMA) guidelines to search, review, analyse and report the findings of the studies included (Figure 1).

### 2.1. Inclusion and Exclusion Criteria

Studies were included if they met all of the following criteria: (1) written in English; (2) published from 2000 up until 3 December 2021; (3) explored at least one of the outcomes of interest of this review which were oral health care knowledge, attitudes and/or practices of people with diabetes in South Asian countries); (4) were conducted among people with any type of diabetes. Non-empirical publications such as letters to the editor, correspondence commentaries, editorials and reviews were excluded. There were no exclusion criteria for age or gender. No restrictions were applied on study setting nor on the study quality.

### 2.2. Outcome Measures

For this review, operational definitions for knowledge, attitudes and practices relating to oral health and diabetes were specified. Outcomes were reported as a percentage, representing a proportion of the total sample. 

Knowledge encompassed the proportion of participants who were aware of the relationship between oral health and diabetes (including oral health complications), the importance of blood glucose management to minimise oral health risks, and sources where individuals received oral health education. 

Attitudes referred to the individual’s perceptions and beliefs towards oral health care, and included perceived barriers to receiving dental care. 

Practices included the behaviours and actions undertaken by individuals to maintain their oral health; specifically, this covered toothbrushing frequency, the proportion of people who used toothpaste or toothpowders, inter-dental cleaning, the use of mouthwash and dental visiting.

### 2.3. Data Sources and Search Strategy

A systematic literature search was carried out in the following databases: Ovid Medline, CINAHL, ProQuest Central, Scopus, Web of Science and Embase. The search was performed using a combination of different keywords and Boolean operators. Appendix A provides the full search strategy used for the review. A hand search of the literature was also conducted, using the reference lists of relevant literature.

### 2.4. Article Selection and Screening

All identified citations were imported into EndNote, software utilised for reference management, and duplicates were removed. Three reviewers (P.P., U.N.Y., A.K.) screened the titles and abstracts independently. Any discrepancies between reviewers during the screening process was resolved through discussion and consensus. The full texts of the remaining articles were reviewed (P.P., U.N.Y., A.K.), and were discussed with another three reviewers for consensus (L.B.R., A.G., M.S.S.).

### 2.5. Data Extraction and Analysis

Relevant data from each of the selected studies were extracted based on the study’s aims. These data included author, year, country, study design, recruitment setting, sample and participant characteristics, blood glucose management, duration since diabetes diagnosis, education levels and findings from the participants’ knowledge, attitudes and practices in relation to oral health and diabetes. Main categories were formed based on the three main domains of oral health knowledge, attitudes and practices, and sub-categories were developed based on the codes identified from within each category.

## 3. Results

A total of 421 articles were identified through the initial search of the six databases and nine articles through other sources. From the initial results, 142 articles were excluded as they were published before 2000 or were duplicates. After screening the titles and abstracts of the papers against the inclusion criteria, a further 254 articles were excluded. A total of 25 articles underwent full-text screening, of which just two were excluded as the samples did not match the eligibility criteria for this review. A total of 23 articles were included in the final review (see Figure 1). 

### 3.1. Demographic Data

There were a total 23 studies from five countries in South Asia: Bangladesh (*n* = 1), India (*n* = 14), Nepal (*n* = 1), Pakistan (*n* = 6), and Sri Lanka (*n* = 1). Most of the studies were cross-sectional (*n* = 21). There was one prospective cohort study and one retrospective analytical case–control study, and participants were recruited primarily from hospitals (*n* = 19). Among the studies that reported on age, all but one study included adults (one study focused on paediatric patients aged 10–15 years). The duration since a diabetes diagnosis varied from 1 month to more than 15 years ago; however, nine studies did not report the duration since diabetes diagnosis. Reporting of educational attainment also varied significantly, from illiterate to achieving post-graduate education. The various categories of education attained varied between studies. Nine studies did not report on education.

Just five studies reported on the blood glucose range of participants, using different instruments to measure this such as glycated haemoglobin (HbA1c); an undescribed categorisation of “good”, “moderate” and “poor” control; and one study reported a random blood sugar test that was within/outside of the target range. Six studies reported the proportion of participants that managed their blood glucose levels through medication. The summary of the included studies with main results are provided in Table 1.

### 3.2. Focus Area 1: Knowledge and Education about Oral Health for Diabetes (n = 21)

There were 21 studies that were included in this focus area [37,38,39,40,41,42,43,44,45,46,47,48,49,50,51,54,55,56,57,58,59]. These studies explored the proportion of participants who had some awareness about diabetes and oral health, and where the participants obtained their oral health knowledge.

#### 3.2.1. Awareness of the Relationship between Oral Health and Diabetes

Fifteen studies reported on whether participants had an awareness of the relationship between diabetes and oral health [37,38,39,40,41,44,46,47,49,50,51,54,56,57,58]. Overall, awareness among participants was low as only 11–64% were aware of this relationship [37,39,41,44,46,47,49,54,56]. Awareness between diabetes and specific conditions appeared to vary. Twelve studies examined the awareness of participants around the relationship between diabetes and a range of conditions and symptoms including periodontal diseases, dental caries, swollen gums, altered wound healing, oral fungal infections, oral malodour, loosening teeth and dryness of mouth [38,42,46,47,48,50,51,54,56,57,58,59].

Four studies, conducted in India and Sri Lanka, reported on oral health knowledge scores, and found that participants had generally lower scores [43,45,55,57]. Three studies categorised knowledge scores based on various cut-offs of “poor”, “fair/moderate”, and “good”. Based on these various questionnaires and scores, there was significant variation in score distributions between two studies (9–37% scored poor, 31–87% scored fair/moderate, and 10–32% scored good) [45,57], and the third study found that participants scored poorly on average [55]. Only one study, by Geetha, Pramila, Jain and Suresh [43], compared knowledge scores among children and adolescents with type 1 diabetes with a control group, and found that those with diabetes had significantly higher oral health knowledge scores.

#### 3.2.2. Sources of Oral Health Education

Eight studies explored where participants had obtained education about diabetes and oral health. These studies were conducted in India (*n* = 5) [38,39,49,51,56] and Pakistan (*n* = 3) [40,46,54]. The participants mostly obtained diabetes and oral health information from dentists (13–63%) and doctors (10–44%). There was one study where none received any information from physicians [39], and another where 1% received information from dentists [46]. Proportionally, fewer participants cited media sources such as television, social media, newspapers and radio (1–39%) and few received information from family and friends (1–12%). Five studies, four of which were conducted in India, found that between 45–98% of respondents had not received any oral health information about oral health or dental referrals [37,38,44,48,57]. 

### 3.3. Focus Area 2: Attitudes towards Perceived Barriers, Education, and Managing Oral Health Problems (n = 15)

In the second focus area, 15 studies were identified [37,38,40,41,43,44,45,46,47,49,51,52,54,56,57]. Two of the studies that used an instrument to measure attitudes found that between 32 and 62% had ‘good’ attitudes towards their oral health [45,57]. The other studies also explored attitudes and receptiveness towards receiving oral health information; and how participants would likely manage their oral health problems, performing oral hygiene self-care; and perceived barriers experienced by participants in accessing dental services.

#### 3.3.1. Receptiveness to More Oral Health Information

Participants were generally receptive to oral health information, and also identified that they would change behaviours if provided with additional education. In two Pakistani studies, 30–45% identified they would increase their brushing frequency if they were provided more information about the relationship between oral health and diabetes [40,54]. In another study, almost 90% reported that they would be more careful about their oral hygiene if they were provided education about its role in their diabetes care [44].

Nine studies explored participants’ attitudes towards obtaining more knowledge about maintaining their oral health. Two studies, in India and Pakistan, found that 93% participants wanted to know more information about diabetes and their oral health [38,47]. Participants in other studies identified various sources of information where they would like to receive information which included the dentist (48–98%), doctor (44–98%), diabetes clinic (82%), printed media (3–100%), radio and TV (44–64%) and the internet (22%) [46,51,56,57]. 

#### 3.3.2. Managing Their Oral Health Problems

Five studies explored participants’ attitudes towards managing their oral health problems [40,44,47,51,54]. Three studies identified that many participants preferred to see a dentist to help manage their oral health problems [47,51,54]. However, in the Pakistani study by Mirza, Khan, Ali and Chaudhry [54], 28% preferred to self-manage their problems although the study did not report on how oral health problems would be managed. In a different study in Sri Lanka, 28% believed that visiting the dentist without any dental symptoms was a waste of money [57]. Further, between 26 and 71% of participants believed that discussing their diabetes status with their dental provider was unimportant [52,57].

#### 3.3.3. Perceived Barriers to Dental Visiting and Oral Hygiene

In five studies, all participants identified the main perceived barriers to dental visits in the previous two years: being asymptomatic (24–68%), lack of awareness (27–30%), time constraints (12–18%), dental fear (3–18%), cost (10–15%), transportation (4%), not being referred, overcrowding and long waiting times, cost to follow up public dentist visits and fatalistic beliefs [37,41,43,44,57].

Only one study, which was conducted in India, explored the perceived barriers for people to maintain personal oral hygiene [49]. Almost half (40%) reported that they did not have enough time; other reasons cited included laziness (31%), lack of awareness (24%), and bleeding gums while brushing (5%).

### 3.4. Focus Area 3: Oral Health Self-Care Practices and Dental Visits (n = 22)

The third focus area, which explored participants’ oral health practices, included 22 studies [37,38,39,40,41,42,43,44,45,46,47,48,49,50,51,52,53,54,55,56,57,58]. These studies explored specific personal oral hygiene practices including toothbrushing frequency, interdental cleaning and use of mouthwashes. Most studies also explored participants’ dental visiting behaviours.

#### 3.4.1. Toothbrushing Frequency

Nineteen studies reported on toothbrushing frequency [37,38,40,41,42,43,44,46,47,49,50,51,52,53,54,55,56,57,58]. Across the studies, there was a large variation (3–93%) of participants who brushed their teeth twice daily; however, 14 studies found that less than 50% brushed their twice daily [37,38,41,42,43,44,46,47,49,50,53,54,56,58]. Two studies that compared a diabetes group to a control found that the diabetes group brushed less frequently compared to the control [43,44].

The proportion of people who brushed their teeth once daily was higher, with 70–100% participants across 17 studies reporting that they brushed at least once daily [37,38,40,41,42,43,44,46,47,49,50,51,52,54,55,56,58]. One Indian study investigated factors that increased the likelihood of participants brushing at least twice daily and this included age (45–54 years), higher levels of education, optimal glycaemic management, and having no diabetes-related complications [37].

Nine studies reported on the proportion of people who used a toothpaste or other alternative tooth-cleaning agent, such as tooth powders, to clean their teeth [37,38,43,44,47,49,51,56,57]. The studies found most participants were using toothpaste (86–100%). 

#### 3.4.2. Interdental Cleaning

Ten studies explored the use of interdental cleaning aids among participants [37,38,40,42,46,47,50,51,53,55]. Across all 10 studies, between 0 and 52% people used interdental cleaning aids. Of these 10 studies, five found that between 0 and 19% of participants had used an inter-dental cleaning aid [38,41,46,53,55]. While only one study specifically reported daily flossing [47], most of the studies just reported whether participants used an inter-dental cleaning aid.

#### 3.4.3. Use of Mouthwashes

Seven studies explored the use of mouthwash, which varied from 18 to 74% (generally higher than the rates of interdental cleaning) [37,38,43,47,50,51,53]. Sabeen Masood, Khan and Sarfaraz [53] found that 5% of participants used a mouthwash daily.

#### 3.4.4. Dental Visiting

Nineteen studies reported on dental visiting [37,38,39,41,42,43,44,46,47,48,49,50,51,52,53,55,56,57,58]. Six studies reported whether participants visited the dentist in the last 12 months, which ranged from 16–76% [37,38,41,42,44,51]. Another seven studies reported whether participants visited the dentist “regularly”, often defined as every 6 or 12 months, which ranged between 0–56% [46,48,49,52,53,57,58].

Fourteen studies explored reasons for dental visits, and found that 28–100% of participants visited the dentist to manage oral health problems or symptoms [37,38,39,41,42,43,46,47,48,49,52,53,56,58]. Eight studies found that few people visited the dentist for a check-up or clean [37,38,39,41,42,43,49,56]. While this varied from 0 to 69%, most studies (*n* = 6/8) reported that less than 25% of participants visited for a check-up or clean [37,38,39,41,42,49]. 

## 4. Discussion

This scoping review synthesised the evidence on the oral health knowledge, attitudes and practices among people with diabetes in South Asian countries, a region where the incidence of diabetes is one of the fastest growing in the world [2]. The evidence from this review highlighted a general trend among individuals towards seeking more knowledge about oral health care. The review also identified how people with diabetes chose to manage their oral health, particularly through self-care oral hygiene practices, in South Asian countries. Based on the literature, this section outlines policy implications of the findings presented in this study and some recommendations to address the gaps identified in this review.

A large proportion of individuals from various studies seemed to have a low awareness about the relationship between oral health and diabetes. Moreover, a few studies highlighted that between half to almost all people with diabetes in their samples did not receive any information about their oral health [37,38,44,48,57]. The paucity in oral health knowledge within this population is unsurprising given that global reviews have found that people with chronic, systemic conditions, including diabetes, have a generally lower knowledge about oral health problems associated with their condition [29,60]. This has also been identified in high-income country, such as Australia [61], where there is universal healthcare coverage. Tuncer and Darby [62] identified that participants in Australia had poor knowledge about the potential oral and periodontal complications associated with diabetes. These studies demonstrate that the dissemination of oral health promotion education and advice among people with diabetes is an area of health need both in developing and developed countries.

In tandem with the low levels of oral health advice and information received, some studies identified that participants were receptive to receiving more oral health promotion information from health care providers, primarily doctors and dentists. In a Saudi Arabian study, Mian et al. [63] similarly found that up to 81% of people who had diabetes had never spoken to their doctor or dentist about oral health. An Australian survey also identified that 90% of people with diabetes did not receive oral health information [61] suggesting that the role of oral health promotion among non-dental care providers appears to be limited in other regions as well. Providing individuals with appropriate oral health promotion information is important as it could improve knowledge about oral health. The study by Poudel, Griffiths, Arora, Wong, Flack, Barker and George [61] also found that among the minority who had received oral health information, they were more likely to have greater oral health knowledge.

This review identified the various attitudes and subsequent strategies that people with diabetes adopted to manage their oral health. Many people reported that they managed their oral health through various self-care oral hygiene practices, primarily through toothbrushing. While most individuals across the review brushed their teeth once daily, toothbrushing twice daily was low compared to populations with diabetes in other regions, such as in Australia [61], Italy [64] and USA [65]. Banyai et al. [66] collected information about oral health behaviours among people with diabetes across 60 countries, with half of respondents originating from Europe and none from South Asia. The survey found that 71% of respondents brushed their teeth twice daily [66], which contrasts to the findings in this review, where less than 50% of people brushed their teeth twice daily in most of the included studies. While most studies did not explore why rates of toothbrushing or inter-dental cleaning were relatively lower, one study in this review found that socioeconomic factors, such as age and higher levels of education, were associated with more frequent self-care oral hygiene practices in addition to optimal glucose management [37]. This highlights the need for targeted oral health interventions for people with diabetes at greater risk of poor oral health.

Despite the significant variation on access to dental services, many individuals only accessed dental care to manage symptomatic oral health problems, rather than for routine check-ups. Non-routine dental visiting is a common phenomenon across other regions, including in more developed countries. One Australian study by Armfield and Ketting [67] found that dental avoidance was predicted by difficulties in paying the dental bill, having no or little trust in their last visited dentist, perceived treatment need, and dental anxiety [67]. A review by Gambhir et al. [68] explored various factors that affected dental services utilisation in India, and identified various consumer and provider issues. While Gambhir et al., Gambhir, Brar, Singh, Sofat and Kakar [68] similarly identified all of the factors from the Australian study [67], the authors also found other factors such as geographical region, community type and ratio of people to dentist to be additional barriers in India. The complexity and interaction of individual and broader systemic issues highlighted by Gambhir, Brar, Singh, Sofat and Kakar [68] could further explain why this current review found that some people only accessed the dentist where there was a perceived dental problem, or preferred to self-manage their oral health problems. However, as Gambhir, Brar, Singh, Sofat and Kakar [68] focused on the challenges in India, further research would need to be conducted to understand whether similar issues would be found across other countries in the South Asian region.

The findings from this review highlight a need to implement evidence-based strategies to improve access to dental care, and also the dissemination of oral health information and advice to people with diabetes in South Asian countries [69]. Bastani et al. [70] identified that since social, environmental and economic determinants play a major role in access to timely dental care, policymakers should focus on reallocating resources to mitigate oral health service access inequality by focusing on the implementation of public health insurance packages and equitable distribution of dental clinics. This review further highlights that people with diabetes in South Asian countries would be a population that may require more targeted policies to enhance their accessibility to timely dental care. Another strategy that could enhance oral health awareness in the context of diabetes management is through an integrated oral health care approach. Since this review reported relatively low rates of routine dental visiting among participants, training non-dental care providers to provide relevant oral health promotion information to clients could be an effective strategy to improve awareness [24]. This strategy could be particularly effective as most individuals from a few studies were receptive to receiving more oral health information, including from doctors and in diabetes clinics. While oral health education directly provided by health care providers was the preferred source of information, developing resources could also supplement the information provided by non-dental care providers. Developing resources or training programs would also need to account for socio-demographic and environmental variations such as language, culture and available access to dental services [60]. Furthermore, to support the capacity building of non-dental care providers in oral health, guidelines would need to be developed to emphasise their potential role, and further research would need to be conducted to explore how these programs and resources should be developed. Oral health is not commonly discussed as part of diabetes care. Furthermore, there is gap of polices and practice guidelines to encourage and assist non-dental care professionals in promoting oral health. Therefore, South Asian countries need to focus on developing policies and practice guidelines to promote oral health among people with diabetes [24,61].

The review findings had some limitations. Firstly, the review included the published literature in English language which had the potential to exclude studies or grey literature published in other languages that focus on the South Asian region. More than half of the studies included were also conducted in India, which may skew the perspective and needs across the diverse South Asian region. Finally, virtually all studies were conducted in a hospital setting which may increase the likelihood of limiting the focus of this review to people who have more significant oral health problems that require dental treatment, which would limit the diversity in the diabetes population represented in this review.

Despite the important limitations, this review provides guidance and options to policy makers and program managers to develop and implement policies, strategies and guidelines to strengthen health care systems that prioritize improving knowledge, attitude and practices on oral health among people with diabetes. It is important that the South Asian countries emphasize establishing integrated care for oral health, allocating adequate resources for oral health care, developing capacity of non-dental care professionals, developing oral health promotional materials, and improving access to and utilization of oral health services. Given that oral health has not yet been an important and integral part of diabetes care in South Asian countries, this review highlights the importance of developing guidelines for mandatory oral health screening of all newly diagnosed people with diabetes and designing and implementing community based integrated approaches to improving health behaviour for promoting oral health and management of diabetes.

## 5. Conclusions

The findings of this review highlighted that among people with diabetes in South Asian countries, there appear to be gaps in oral health knowledge, attitudes, and practices, which warrants a need for enhancing the oral health awareness through education, particularly by non-dental care providers. Some people with diabetes reported engaging in oral health practices such as toothbrushing and dental visiting; however, the findings indicated that these could be improved through targeted oral health interventions. Strategies that could improve oral health awareness, attitudes and practices would need to focus on addressing the issue of poor access and high cost of dental treatment through the targeted policies and provisions of oral acre. There is also a need for developing and implementing integrated oral health care approach that would enhance the capacity of non-dental care professionals to promote oral health among patients. Future research would need to develop and evaluate the effectiveness, feasibility and acceptability of these models of integrated oral care within the South Asian countries.

## Figures and Tables

**Figure 1 ijerph-19-13851-f001:**
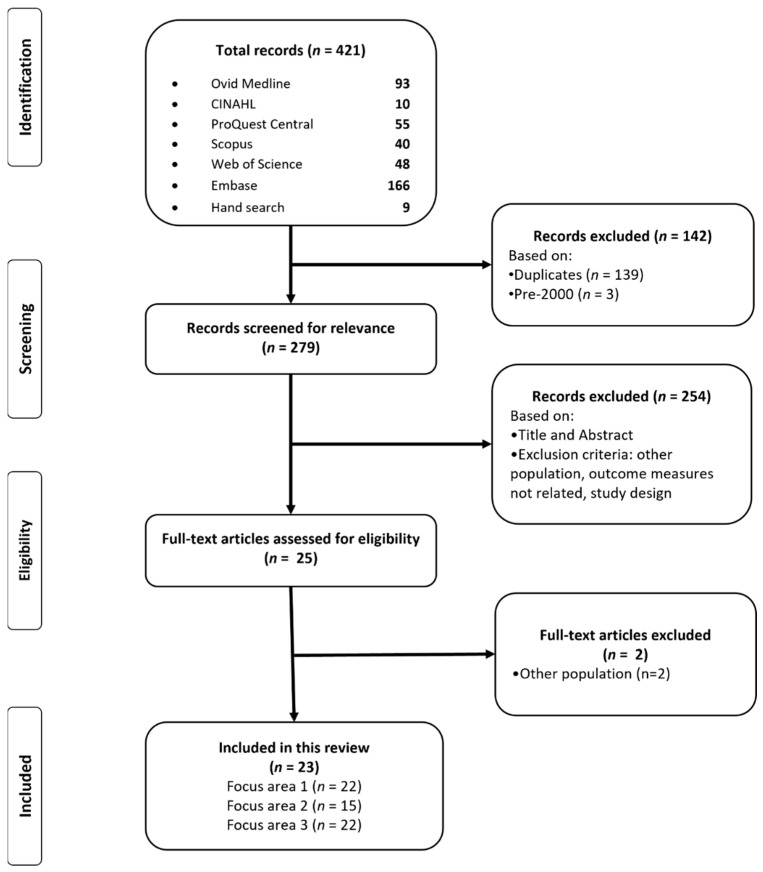
PRISMA flow chart of the study selection process.

**Table 1 ijerph-19-13851-t001:** Summary of the included studies with main results.

Author (Year) Country	Study Design	Setting and Participants	Education	Blood Glucose Management	Duration since Diagnosis	Findings		
						Knowledge (% of People)	Attitudes (% of People)	Practices (% of People)
Aggarwal et al. (2012) [37] India	Cross-sectional	Hospital (oral medicine and radiology) *n* = 500 Age range: 35–87 Male: 53% Female: 47%	Elementary: 32.4% College: 48.2% Post-graduate: 19.4%	HbA1c: <6.0: 56% 6.1–7.5: 28.2% >7.5: 16%	<2 y: 23% 2–5 y: 34% 6–10 y: 26% >10 y: 17%	Aware of the relationship between diabetes and oral health (38%)	Receptive to more information NR Attitudes to manage oral health problems NR Perceived barriers to dental visits: lack of awareness (30%), lack of dental problems (24%), dental fear (18%), cost (15%), and lack of time (12%)	Brush 2× daily (33.4%) Brush 1× daily (49%) Toothpaste (92.4%) Interdental cleaning (52%) Mouthwash (18%) Visit dentist in the last 12 m (76%)
Anitha et al. (2017) [38] India	Cross-sectional	Hospital *n* = 200 Male: 51.5% Female: 48.5%	Educated: 51% Uneducated: 49%	Blood glucose control Good: 34% Moderate: 33% Poor: 34%	<2 y: 10% 2–5 y: 33% >5 y: 58%	Aware of the relationship between diabetes and oral health (30%) Aware of the relationship between diabetes and: swollen gums (43%); oral malodour (63%); loose teeth (22%); altered wound healing (8%); bleeding gums (11%)	Receptive to more information (93%) Attitudes to manage oral health problems NR Perceived barriers NR	Brush 2× daily (3%) Brush 1× daily (97%) Toothpaste (89%) Interdental cleaning (15%) Mouthwash (74%) Visit dentist in the last 12 m (48%)
Arunkumar et al. (2015) [39] India	Cross-sectional	Hospital (outpatient, general medicine) *n* = 185 Age mean: 57.7 y Male: 58% Female: 42%	Illiterate: 9% Dropout: 52% Educated: 39%	Random blood sugar Within target range: 57% Outside target range: 43%	Mean 8.82 y	Aware of the relationship between diabetes and oral health (11%)	Receptive to more information NR Attitudes to manage oral health problems NR Perceived barriers NR	Brush 2× daily NR Brush 1× daily NR Toothpaste NR Interdental cleaning NR Mouthwash NR Visit dentist in the last 12 m NR
Bangash et al. (2011) [40] Pakistan	Cross-sectional	Hospital (dental) *n* = 300 Age mean: 49 y Male: 65% Female: 35%	NR	NR	NR	Aware of the relationship between diabetes and oral health (64%)	Receptive to more information (80%) Attitudes to manage oral health problems NR Perceived barriers NR	Brush 2× daily (86%) Brush 1× daily (14%) Toothpaste NR Interdental cleaning NR Mouthwash NR Visit dentist in the last 12 m NR
Basu et al. (2020) [41] India	Cross-sectional	Hospital (diabetes) *n* = 339 Age range: 35–55 Age mean: 46.1 y Male: 57% Female: 43%	<10 y education: 71% ≥10 y education: 29%	Glycaemic management: Optimal: 32% Suboptimal: 68%	Mean 6 y ≤5 y: 65% ≥6 y: 35%	Aware of the relationship between diabetes and oral health (15%)	Receptive to more information NR Attitudes to manage oral health problems NR Perceived barriers to dental visits: being asymptomatic, not being referred, overcrowding and long waiting times at public health clinics, cost of follow-up visits, fatalistic beliefs (% not reported)	Brush 2× daily (19%) Brush 1× daily (79%) Toothpaste NR Interdental cleaning (5%) Mouthwash NR Visit dentist in the last 12 m (16%)
Champatyray et al. (2020) [42] India	Cross-sectional	Hospital (diabetes in oral pathology) *n* = 100 Age mean: 35 y Male: 59% Female: 39%	NR	NR	NR	Aware of the relationship between diabetes and oral health: NR Aware of the relationship between diabetes and: periodontal disease (60%), dental caries (54%), oral fungal infections (42%)	Receptive to more information NR Attitudes to manage oral health problems NR Perceived barriers NR	Brush 2× daily (49%) Brush 1× daily (43%) Toothpaste NR Interdental cleaning (49%) Mouthwash NR Visit dentist in the last 12 m (60%)
Geetha et al. (2019) [43] India	Retrospective analytical case–control	Hospital (diabetes) *n* = 175 (control *n* = 175) Age mean: 12.9 y Range 10–15 Male: 42% Female: 58%	NR	NR	NR	Aware of the relationship between diabetes and oral health: NR Oral health knowledge scores were higher for people with diabetes (mean 8.3/11) than control group (mean 7.5/11); *p* = 0.001.	Receptive to more information NR Attitudes to manage oral health problems NR Perceived barriers to dental visits: lack of pain (55%), lack of awareness (27%), high cost (10%), fear/anxiety (5%), and transportation problems (4%)	Brush 2× daily (36%) Brush 1× daily (64%) Toothpaste (100%) Interdental cleaning NR Mouthwash (63%) Visit dentist in the last 12 m NR
Gupta et al. (2017) [44] India	Cross-sectional	Hospital (endocrinology) *n* = 100 (control *n* = 100) Age range: 30–70 Male: 56% Female: 44%	None: 4% Matriculate: 45% ≥Graduate: 51%	Medication: 89%	1 m: 3% 6 m: 3% 1 y: 4% >1 y: 90%	Aware of the relationship between diabetes and oral health (24%)	Receptive to more information (88%) Attitudes to manage oral health problems NR Perceived barriers to dental visits: lack of symptoms (68%), lack of time (18%), fear (3%) and cost (11%)	Brush 2× daily (38%) Brush 1× daily (51%) Toothpaste (97.5%) Interdental cleaning NR Mouthwash NR Visit dentist in the last 12 m (50%)
Hasan et al. (2021) [45] Bangladesh	Prospective cohort	Hospital (outpatient, diabetes) *n* = 379 Age range: 21–59 Male: 46% Female: 54%	None: 13% Primary: 34% Secondary: 26% >Secondary: 27%	HbA1c: <7%: 27% ≥7%: 73%	<5 y: 52% ≥5 y: 48%	Aware of the relationship between diabetes and oral health: NR Knowledge score: poor (37%), fair (31%), good (32%)	Oral health attitude score: poor (34%), fair (34%), good (32%) Receptive to more information NR Attitudes to manage oral health problems NR Perceived barriers NR	Oral hygiene practice score: poor (37%), fair (28%), good (35%) Brush 2× daily NR Brush 1× daily NR Toothpaste NR Interdental cleaning NR Mouthwash NR Visit dentist in the last 12 m NR
Islam et al. (2021) [46] Pakistan	Cross-sectional	Hospital (diabetes) *n* = 400 Age mean: 51 y Male: NR Female: NR (majority female)	Educational level NR	NR	Mean 3 y	Aware of the relationship between diabetes and oral health (60%) Aware of the relationship between diabetes and: periodontal disease (11%), dental caries (2%)	Receptive to more information (21%) Attitudes to manage oral health problems NR Perceived barriers NR	Brush 2× daily (47%) Brush 1× daily (52%) Toothpaste NR Interdental cleaning (19%) Mouthwash NR Visit dentist in the last 12 m NR
Javaid et al. (2019) [47] Pakistan	Cross-sectional	Military hospital *n* = 344 Male: 38% Female: 62%	No school: 11% <5 y: 12% <10 y: 22% >10 y: 55%	Medication Oral: 58% Insulin: 20% Self-remedy/herb: 10% Insulin and oral: 13%	<5 y: 40% >5–10 y: 19% >10 y: 41%	Aware of the relationship between diabetes and oral health (64%) Aware of relationship between diabetes and: periodontitis (10%), caries and gingivitis (70%)	Receptive to oral health education (93%) Attitudes to manage oral health problems: ‘majority’ prefer to visit a dentist to manage dental problem Perceived barriers NR	Brush 2× daily (38%) Brush 1× daily (48%) Toothpaste (92.2%) Interdental cleaning (31%) Mouthwash (19%) Visit dentist in the last 12 m NR
Jayan et al. (2020) [48] India	Cross-sectional	Hospital (dental college) *n* = 302 Male: NR Female: NR	Graduate: 64.8%	NR	2–5 y: 37% >10 y: 40%	Aware of the relationship between diabetes and oral health: NR Aware of relationship between diabetes and delayed wound dealing (70%), mouth dryness (48%)	Receptive to more information NR Attitudes to manage oral health problems NR Perceived barriers NR	Brush 2× daily NR Brush 1× daily NR Toothpaste NR Interdental cleaning NR Mouthwash NR Visits dentist every 6 m (20%) Does not see dentist (42%)
Kamalli et al. (2020) [49] India	Cross-sectional	Setting NR *n* = 87 Male: NR Female: NR	School: 29% Graduate: 49% Post-graduate: 22%	NR	NR	Aware of the relationship between diabetes and oral health: NR	Receptive to more information NR Attitudes to manage oral health problems NR Perceived barriers to maintaining oral hygiene: insufficient time (40%), laziness (31%), lack of awareness of oral hygiene (24.1%) and bleeding of gums while brushing (5%)	Brush 2× daily (17%) Brush 1× daily (83%) Toothpaste (86.2%) Interdental cleaning NR Mouthwash NR Visits dentist every 6–12 m (10.3%)
Kamath et al. (2015) [50] India	Cross-sectional	Hospital (diabetes and outpatient, periodontology department) *n* = 138 Male: 49% Female: 51%	NR	Mean HbA1c Males: 6.72 (SD 0.68) Females: 6.81 (SD 0.76)	1–10 y: 62% 11–20 y: 30% 21–30 y: 8%	Aware of the relationship between diabetes and oral health (52%) Aware of the relationship between diabetes and dental caries, periodontal disease and dry mouth (29–72%)	Receptive to more information NR Attitudes to manage oral health problems NR Perceived barriers NR	Brush 2× daily (33%) Brush 1× daily (37%) Toothpaste (86.2%) Interdental cleaning (35%) Mouthwash (55%) Visit dentist in the last 6 m (28%)
Kejriwal et al. (2014) [51] India	Cross-sectional	Hospital (dental) *n* = 300 Age range NR Male: 67% Female: 33%	NR	NR	<4 y: 53% 5–10 y: 25% 11–20 y: 14% >20 y: 8%	Aware of the relationship between diabetes and oral health (50%) Aware of relationship between diabetes and: dryness of mouth (18%), bleeding gums (16%), tooth decay (14%), soreness (5%), ulcers (4%), infections (1%), all of the above (12%)	Receptive to more information NR Attitudes to manage oral health problems: prefers to see dentist (57%), doctor (41%), self-manage (2%), ignore (6%) Perceived barriers NR	Brush 2× daily (52%) Brush 1× daily (35%) Toothpaste (98.7%) Interdental cleaning (24%) Mouthwash (52%) Visit dentist in the last 12 m (45%)
Mainali et al. (2013) [52] Nepal	Cross-sectional	Hospital and dental college *n* = 100 Male: NR Female: NR	NR	Medication Oral: 81% Insulin: 5% Insulin and oral: 1% None: 13%	NR	Aware of the relationship between diabetes and oral health: NR	Receptive to more information NR Attitudes to manage oral health problems: 26% believed that dentists did not need to know about diabetes status Perceived barriers NR	Brush 2× daily (79%) Brush 1× daily (21%) Toothpaste NR Interdental cleaning NR Mouthwash NR Visit dentist every 6–12 m (56%)
Masood et al. (2019) [53] Pakistan	Cross-sectional	Hospital (dental) *n* = 120 Male: 36% Female: 64%	NR	NR	NR	Aware of the relationship between diabetes and oral health: NR	Receptive to more information NR Attitudes to manage oral health problems NR Perceived barriers NR	Brush 2× daily (30%) Brush 1× daily (45%) Toothpaste NR Interdental cleaning (0%) Mouthwash (5%) Visit dentist regularly (36%)
Mirza et al. (2007) [54] Pakistan	Cross-sectional	Hospital (diabetes clinic) *n* = 240 Age mean: 49 Age range: 17–80 Male: 42% Female: 58%	NR	NR	NR	Aware of the relationship between diabetes and oral health (43%) and oral complications (35%)	Receptive to more information: 68% (increase toothbrushing frequency (45%), consult a dentist (23%), make no changes (31.5%)) Attitudes to manage oral health problems: prefers to see dentist (47%), self-remedy (28%) Perceived barriers NR	Brush 2× daily (24%) Brush 1× daily NR Toothpaste NR Interdental cleaning NR Mouthwash NR Visit dentist in the last 12 m NR
Parakh et al. (2020) [55] India	Cross-sectional	Hospital (dental college)*n* = 447Age range: 25–60Male: 54%Female: 46%	Illiterate: 3% Primary: 9%Middle: 43%≥Higher secondary: 45%	Managing diabetes with medication: 98%	< 1 year: 5%1–3 y: 18%3–5 y: 35%>5 y: 43%	Aware of the relationship between diabetes and oral health: NR Knowledge score about oral manifestations: poor (avg 4.92/12)	Receptive to more information NR Attitudes to manage oral health problems NR Perceived barriers NR	Brush 2× daily (93%) Brush 1× daily (7%) Toothpaste NR Interdental cleaning (0%) Mouthwash NR Visit dentist in the last 6–12 m (60%)
Shanmukappa et al. (2017) [56] India	Cross-sectional	Diabetes centres, private dental clinics and hospital (dental college)*n* = 600Age range NR (41–50 y: 37%)Male: NRFemale: NR	>10th grade: 54%	Managing diabetes with medication: 88%	1–5 y: 36%	Aware of the relationship between diabetes and oral health (24%) Aware of the relationship between diabetes and gum disease (31%)	Receptive to more information NR Attitudes to manage oral health problems NR Perceived barriers NR	Brush 2× daily (28%) Brush 1× daily (70%) Toothpaste (94%) Interdental cleaning NR Mouthwash NR Visit dentist every 6m (19%)
Silva et al. (2016) [57] Sri Lanka	Cross-sectional	Hospital *n* = 427 Age range: 18–73 (83% >50)Male: 26%Female: 74%	< GCE: 48%Up to GCE: 36%> GCE: 16%	NR	<5 y: 41%>15 y: 17%	Aware of the relationship between diabetes and oral health (56%) Oral health knowledge score: poor (9%), moderate (87%), good (10%) Aware of relationship between diabetes and: wound healing (89%), gingivitis (93%) and oral thrush (15%)	Oral hygiene attitude score: poor (4%), moderate (34%), good (62%) Receptive to oral health information at diabetes clinic (82%) Attitudes to manage oral health problems: visiting the dentist without oral health problems was a waste of money (28%); discussing diabetes with dentist unimportant (71%) Perceived barriers NR	Oral hygiene practice score: poor (4%), moderate (40%), good (56%) Brush 2× daily (92%) Brush 1× daily NR Toothpaste (96.25%) Interdental cleaning NR Mouthwash NR Visits dentist every 6 m (0.2%)
Sriram et al. (2020) [58] India	Cross-sectional	Hospital (dental) *n* = 100 Age range: 30–60 Male: 60% Female: 40%	Illiterate: 22% School: 33% Graduate: 45%	NR	NR	Aware of the relationship between diabetes and oral health: NR Aware of the relationship between diabetes and: gum problems (20%), oral/systemic problems (37%), dry mouth (30%), fungal disease (10%), caries (40%)	Receptive to more information NR Attitudes to manage oral health problems NR Perceived barriers NR	Brush 2× daily (24%) Brush 1× daily (67%) Toothpaste NR Interdental cleaning (%) Mouthwash NR Visit dentist every 6–12 m (55%)
Talpur et al. (2015) [59] Pakistan	Cross-sectional	Hospital (diabetes clinic) *n* = 200 Age mean: 50 y Male: 49% Female: 52%	NR	NR	NR	Aware of the relationship between diabetes and oral health: NR Aware of the relationship between diabetes and: gingivitis (49.2%), tooth loss (27.5%), periodontal disease (16.4%), and oral infections (6.9%)	Receptive to more information NR Attitudes to manage oral health problems NR Perceived barriers NR	Brush 2× daily NR Brush 1× daily NR Toothpaste NR Interdental cleaning NR Mouthwash NR Visit dentist in the last 12 m NR

GCE: General Certificate of Education; HbA1c: glycosylated haemoglobin; m: months; NR: not reported; y: years.

## Data Availability

Not applicable.

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
