# Peer review of "Oral Health Knowledge, Attitudes and Practices of People Living with Diabetes in South Asia: A Scoping Review"

_ijerph, 2022, doi:10.3390/ijerph192113851_

Round 1
Reviewer 1 Report
I read the manuscript with interest.
Indeed, this is a well-conducted and reported scoping review. It is hard to believe that 8 authors would be involved in such an activity. This number is rather excessive. Anyway, I am glad the credits of each author have been addressed.
The following reference can be cited in the discussion when addressing the level of evidence.
Saudi Dent J. 2017 Jul;29(3):83-92. doi: 10.1016/j.sdentj.2017.02.002. Epub 2017 Mar 15.
Author Response
Comment 1
I read the manuscript with interest.
Indeed, this is a well-conducted and reported scoping review. It is hard to believe that 8 authors would be involved in such an activity. This number is rather excessive. Anyway, I am glad the credits of each author have been addressed.
The following reference can be cited in the discussion when addressing the level of evidence: Saudi Dent J. 2017 Jul;29(3):83-92. doi: 10.1016/j.sdentj.2017.02.002. Epub 2017 Mar 15.
Response 1
We have included this reference in the Discussion (line 407, reference 69).
Reviewer 2 Report
Most of the reviewed papers are from the same country.
Author Response
Comment 1
Most of the reviewed papers are from the same country.
Response 1
The search employed a comprehensive and sensitive strategy to cover a range of countries within the South Asian region. We would also like to highlight that of the 23 studies included, 14 were from India but the remaining 9 were from other countries (lines 177-179).
In the limitations section of the Discussion we have changed “Most of the studies…” to “More than half of the studies included” to more accurately reflect the proportion reported (line 435).
Reviewer 3 Report
The manuscript “Oral health knowledge, attitudes and practices of people living with diabetes in South Asia: a scoping review” is significant in the context of oral health hygiene among the people living with NCDs. I have suggested the following points to consider.
In Methods clearly define your outcome variables. The outcome must be in percentage or in number.
“Attitudes referred to the individual’s perceptions and beliefs towards oral health care, and included barriers to receiving dental care.” Barriers are not considered attitudes. Make a separate heading for barriers.
In table 1, the authors need to organise the findings.
Table 1. Summary of the included studies with main results. There is confusion between knowledge, attitude, and practices in many studies.
Use Landscape Table format.
Author (year); Country Study; Design; Setting, Participants; Education; Sample size; Blood glucose management; Duration since diagnosis; Knowledge; Attitude; Practice; Barriers
Under knowledge present (% of people aware of the relationship between diabetes and oral health)
Under Attitude (% have positive belief towards oral health care and hygiene)
Practices (% of people brush their teeth twice daily, % of people brush their teeth at least once daily; % of people use any type of toothpaste).
Under Barriers (% of people face difficulty in seeking care although they need it).
The remaining content describes in the text under the heading Knowledge; Attitude; Practice; Barriers.
If any information under the Knowledge; Attitude; Practice; Barriers section is presented by at least five studies try to find out the pooled prevalence using meta-analysis (forest plot).
In the discussion section Implication for the policy and practice section is missing.
Author Response
Comment 1
The manuscript “Oral health knowledge, attitudes and practices of people living with diabetes in South Asia: a scoping review” is significant in the context of oral health hygiene among the people living with NCDs. I have suggested the following points to consider.
Response 1
Thank you for taking the time to review this paper.
Comment 2
In Methods clearly define your outcome variables. The outcome must be in percentage or in number.
Response 2
We have identified outcome measures as a separate subheading (2.2. Outcome measures). We have also specified that outcomes will be reported in percentages which represent a proportion of the total sample (line 128-132).
Comment 3
“Attitudes referred to the individual’s perceptions and beliefs towards oral health care, and included barriers to receiving dental care.” Barriers are not considered attitudes. Make a separate heading for barriers.
Response 3
We have edited this to read: Attitudes referred to the individual’s perceptions and beliefs towards oral health care, and included perceived barriers to receiving dental care. (line 137-138)
Comment 4
In table 1, the authors need to organise the findings.
Response 4
Table 1 has been revised as discussed in the subsequent responses.
Comment 5
Table 1. Summary of the included studies with main results. There is confusion between knowledge, attitude, and practices in many studies.
Response 5
The categories (knowledge, attitudes and practices) are consistent with our definition of the categories in the methods (see 2.2. Outcome measures).
Comment 6
Use Landscape Table format.
Response 6
The table has been changed to a landscape format.
Comment 7
Author (year) Country Study; Design; Setting, Participants; Education; Sample size; Blood glucose management; Duration since diagnosis; Knowledge; Attitude; Practice; Barriers
Response 7
We have edited the headings to: Author (year) & Country; Study Design; Setting & Participants; Education; Blood glucose management; Duration since diagnosis; Knowledge; Attitude; Practices (all under ‘Findings’).
We included the sample size under ‘Setting and Participants’ due to page size limitations. We have included ‘perceived barriers’ in ‘Attitudes’.
Comment 8
Under knowledge present (% of people aware of the relationship between diabetes and oral health)
Response 8
In the Results section (Section 3.2.1. Awareness of the relationship between oral health and diabetes), we have specified the % of participants’ awareness of the relationship between diabetes and oral health. (line 203-205).
Comment 9
Under Attitude (% have positive belief towards oral health care and hygiene)
Response 9
We have included a % of participants who had ‘good’ beliefs towards their oral health based on two studies that used an instrument that included a measure of attitudes (line 236-237). More information about their beliefs (reported as a %) has been detailed in the subsections (line 254-294).
Comment 10
Practices (% of people brush their teeth twice daily, % of people brush their teeth at least once daily; % of people use any type of toothpaste).
Response 10
We have highlighted the percentage of people who brush their teeth twice daily (line 302-304), and once daily (line 307-308). The percentage of people who use any type of toothpaste was not part of the outcome measures initially identified (line 139-141).
Comment 11
Under Barriers (% of people face difficulty in seeking care although they need it).
Response 11
As mentioned earlier, perceived barriers have been listed under ‘Attitudes’. We have specified that all participants in five studies identified perceived barriers, and have listed them alongside the proportion of people who reported these barriers (line 285). As some studies were qualitative, the percentage of people who experienced certain barriers were not reported.
Comment 12
The remaining content describes in the text under the heading Knowledge; Attitude; Practice; Barriers.
Response 12
We have reported on Knowledge, Attitudes and Practices, and have detailed the content under their corresponding headings in the Results section.
Comment 13
If any information under the Knowledge; Attitude; Practice; Barriers section is presented by at least five studies try to find out the pooled prevalence using meta-analysis (forest plot).
Response 13
From our understanding, a forest plot is not typically used outside the context of a systematic review or meta-analysis and is more suited to assess the effect of an outcome measure.
- Peters, M. D., Marnie, C., Tricco, A. C., Pollock, D., Munn, Z., Alexander, L., … & Khalil, H. (2020). Updated methodological guidance for the conduct of scoping reviews. JBI evidence synthesis, 18(10), 2119-2126
- Akobeng, A. K. Understanding systematic reviews and meta-analysis. Archives of Disease in Childhood 2005;90:845-848.
Comment 14
In the discussion section Implication for the policy and practice section is missing.
Response 14
Thank you for this important feedback. We have now added following important policy implications to the manuscript. See lines 442-454. This reads as below:
Despite the important limitations, this review provides guidance and options to policy makers and program managers to develop and implement policies, strategies and guidelines to strengthen health care systems that prioritize improving knowledge, attitude and practices on oral health among people with diabetes. It is important that the South Asian countries emphasize establishing integrated care for oral health, allocating adequate resources for oral health care, developing capacity of non-dental care professionals, developing oral health promotional materials, and improving access to and utilization of oral health services. Given that oral health has not yet been an important and integral part of diabetes care in South Asian countries, this review highlights the importance of developing guidelines for mandatory oral health screening of all newly diagnosed people with diabetes and designing and implementing community based integrated approach to improving health behaviour for promoting oral health and management of diabetes.
Round 2
Reviewer 3 Report
Thanks for addressing all the suggestions. However, Table 1 needs to be condensed. The authors not provided content in the table as suggested, particularly Knowledge, Attitude, and Practice. Only present the following content in for knowledge, attitude and practice. The remaining % value, please describe in the content of the results section, not in Table. If any study has not presented mentioned variables, write it as Not provided. Following are the table heading.
% of people aware of the relationship between diabetes and oral health)
% have positive beliefs towards oral health care and hygiene
% of people brush their teeth twice daily,
% of people brush their teeth at least once daily; % of people use any type of toothpaste).
Author Response
Reviewer Comment
Thanks for addressing all the suggestions. However, Table 1 needs to be condensed. The authors not provided content in the table as suggested, particularly Knowledge, Attitude, and Practice. Only present the following content in for knowledge, attitude and practice. The remaining % value, please describe in the content of the results section, not in Table. If any study has not presented mentioned variables, write it as Not provided. Following are the table heading.
% of people aware of the relationship between diabetes and oral health)
% have positive beliefs towards oral health care and hygiene
% of people brush their teeth twice daily,
% of people brush their teeth at least once daily; % of people use any type of toothpaste).
Author Response
Thank you for your suggestion. As suggested, we have condensed Table 1 as follows:
- The knowledge section containing only information about the % of people aware of relationship between diabetes and oral health as well as other specific areas.
- The practice section only contains information around % of people who brush twice a day, once daily, % of people who reported using toothpaste, use interdental cleaning, mouthwash and visit a dentist in the last 12 months. All remaining information have been deleted from Table 1 but left in the results section.
- In relation in Attitudes, because of the wide variation of findings we are unable to restrict the heading to positive beliefs towards oral health care and hygiene. Instead, we have used the following headings: Receptiveness to more information; attitudes towards managing oral health problems; and perceived barriers to dental visits and oral hygiene.
- “Not reported” was used when any study did not present the mentioned variables